# On the Chip Shaping and Surface Topography When Finish Cutting 17-4 PH Precipitation-Hardening Stainless Steel under Near-Dry Cutting Conditions

**DOI:** 10.3390/ma13092188

**Published:** 2020-05-09

**Authors:** Kamil Leksycki, Eugene Feldshtein, Grzegorz M. Królczyk, Stanisław Legutko

**Affiliations:** 1Institute of Mechanical Engineering, University of Zielona Gora, 4 Prof. Z. Szafrana Street, 65-516 Zielona Gora, Poland; k.leksycki@ibem.uz.zgora.pl (K.L.); e.feldsztein@ibem.uz.zgora.pl (E.F.); 2Department of Manufacturing Engineering and Production Automation, Opole University of Technology, 76, Proszkowska Street, 45-758 Opole, Poland; g.krolczyk@po.opole.pl; 3Faculty of Mechanical Engineering, Poznan University of Technology, 3 Piotrowo Street, 60-965 Poznan, Poland

**Keywords:** stainless steel, finish turning, chip shaping, surface topography, cutting condition, near-dry cutting, extreme pressure (EP) addition

## Abstract

This study describes the surface topography of the 17-4 PH stainless steel machined under dry, wet and near-dry cutting conditions. Cutting speeds of 150–500 m/min, feeds of 0.05–0.4 mm/rev and 0.5 mm depth of cutting were applied. The research was based on the ‘parameter space investigation’ method. Surface roughness parameters, contour maps and material participation curves were analysed using the optical Sensofar S Neox 3D profilometer and the effect of feed, cutting speed and their mutual interaction was noticed. Changes in chip shape depending on the processing conditions are shown. Compared to dry machining, a reduction of *Sa*, *Sq* and *Sz* parameters of 38–48% was achieved for near-dry condition. For lower feeds and average cutting speeds valleys and ridges were observed on the surface machined under dry, wet and near-dry conditions. For higher feeds and middle and higher cutting speeds, deep valleys and high ridges were observed on the surface. Depending on the processing conditions, different textures of the machined surface were registered, particularly anisotropic mixed, periodic and periodically determined. In the *Sa* range of 0.4–0.8 μm for dry and wet conditions the surface isotropy is ~20%, under near-dry conditions it is ~60%.

## 1. Introduction

The 17-4 PH steel is a relatively new and increasingly popular stainless material for medical devices [1,2]. In particular, it is widely used in the production of surgical instruments [1,3] and orthopaedic instruments [4].

The chip shape is the result of physical-chemical phenomena occurring in the cutting zone and plays a significant role in the cutting process and the quality of the surface machined. The chip may occupy a volume many times greater than the material from which it was made and may also cause difficulties in its removal from the machining zone [5]. When turning, the cutting edge of the insert is in constant contact with the material being machined, which usually leads forming of long and snarling chips. This chip often wounds around the cutting tool and causes abrasions on the machined surface [6]. Therefore, the use of machining fluids in the cutting process is an important factor influencing the appropriate chip formation and effective removal from the cutting zone [7,8]. One of the main factors determining the effectiveness of the materials use is the topography of machined surface [9]. It has a significant impact on the operating properties of biomedical materials parts, particularly on wear and corrosion resistance [10]. When turning, one of the factors influencing the surface topography is the type of cooling lubricant used, which is also very important for environmental and human health risks [11].

Therefore, current scientific research is focused on the search for improvements in machining processes for sustainable clean production, including minimizing the use of cooling lubricants and improving surface quality [12,13].

Pereira et al. [14] tested the combination of cryogenic and minimum quantity lubrication (MQL) methods while carrying out the ecological processing of the AISI 304 stainless steel. The better results in comparison with other cooling methods were ensured. Zou et al. [15] analysed the roughness and surface topography of the 17-4 PH steel under different processing conditions. The cryogenic cooling provided more favourable results. Sivaiah and Chakradhar [16] examined the cryogenic machining of the 17-4 PH steel. It was found that in comparison to wet cooling, cryogenic cooling reduces surface roughness. Sivaiah and Chakradhar [17] studied the influence of liquid N_2_ cooling when cutting the 17-4 PH steel in comparison with other cooling conditions. At each cutting depth cryogenic turning reduced surface roughness and improved surface integrity in comparison with other methods. Leksycki and Feldshtein [18] investigated the formation of surface texture of the 17-4PH (X5CrNiCuNb16-4) steel under dry and wet turning. In comparison with dry processing, the reduction of roughness parameters about 30% was registered during wet turning. Jerold and Kumar [19] tested the CO_2_ effect as a cooling medium on the surface roughness when cutting the 316 steel. The results were compared with other cooling methods and revealed that CO_2_ increased the surface integrity in the range of 4–52%. Krolczyk et al. [20] presented different images of the surface topography of the 1.4462 steel after dry and minimum quantity cooling lubrication (MQCL) turning. Compared to dry machining, the 3D surface texture parameters were reduced under MQLC conditions when using higher cutting parameters. In comparison to dry machining, a more favourable surface was obtained under MQCL conditions. Maruda et al. [21] tested the effect of different near-dry cutting conditions on *Ra* and *Rq* values of the 1.4310 (X10CrNi18-8) steel. It was observed that in comparison with dry cutting near-dry cutting causes a decrease in the roughness parameters from 2% to 42%. Rajaguru and Arunachalam [22] tested the impact of cooling conditions on the topography and surface roughness profiles of a super duplex steel. MQL machining provided better results. Elmunafi et al. [23] evaluated the effectiveness of MQL machining with a castor oil when turning the AISI 420 steel. Compared to dry machining conditions, the MQL conditions resulted in a better tool life, but a slight increase of the surface roughness was observed.

Summarizing, it can be said that in the current state of knowledge the possibilities of modern near-dry cutting methods in the machining of stainless steels are presented quite extensively, however, there is a lack of analysis of the chip shaping and three dimensional surface topography of modern precipitation-hardening (PH) steels. The main aim of this research is the comparison of different near-dry conditions efficiencies on the chip shaping and surface texture after finish cutting of the 17-4 PH steel.

## 2. Materials and Methods

The 17-4 PH (AISI 630) steel was machined, whose chemical composition and mechanical properties correspond to ASTM A693 Grade 630 standard.

The 17-4 PH steel is precipitation-hardening chromium-nickel steel with a martensitic structure containing a small volume of austenite. The steel contains copper (about 3%) and is reinforced with copper-rich particulates because of their precipitation from the metal matrix [24,25]. According to DIN EN 10088-1:2014-12, the material has a modulus of elasticity of 195 GPa, a tensile strength of 930–1100 MPa and a yield strength of 720 MPa. The steel has the excellent corrosion resistance in comparison with typical 304 or 316L stainless steels [2], while it has low thermal conductivity, which reduces the machinability and negatively affects the surface quality and the life of cutter [26].

The tests were carried out using the CTX 510 machining centre (DMG MORI, Pleszew, Poland). A turning tool was used with CoroTurn SDJCR 2525M 11 tool holder (Sandvik Coromant, Sandviken, Sweden) and CoroTurn DCMX 11 T3 04-WM 1115 insert (Sandvik Coromant, Sandviken, Sweden). The cutting material was GC 1115 cemented carbide (group S according to ISO 513:2012 standard) with PVD coating.

Turning was performed with cutting speeds *v_c_* = 150–500 m/min, feeds *f* = 0.05–0.4 mm/rev with the 0.5 mm depth of cutting that corresponds with the finish machining conditions. The machining was carried out using samples with the length of 15 mm the diameter of 50 mm. Three measurements were made for each surface treated.

Following cooling conditions were applied: dry, wet, minimum quantity lubrication (MQL) and MQL with extreme pressure (EP) addition (extreme pressure additive based on phosphorus ester). EP additive is usually added to protect the surface against wear [27,28]. A water emulsion of Castrol Alusol SL 51 XBB emulsifying oil with 7% concentration was used as a cutting fluid. Oil mist was produced as a mixture of air and ECOCUT MIKRO 20 E oil. Phosphorus ester was used as an extreme pressure additive. Such additives are used to decrease wear of the parts that work under very high pressures. They are also added to cutting fluids for machining of metals.

Lenox 1LN Micronizer (Lenox, East Longmeadow, MA, USA) was used to generate oil mist. Constant air and oil flows as well as constant air pressure were applied, which ensure the best conditions for mist production [28,29].

Topography parameters of the surface of the machined steel were measured with the use of Sensofar S Neox 3D optical profilometer (Sensofar Group, Barcelona, Spain) using Confocal Profilers technology. The measurement results were prepared using Mountains Maps Premium 7.4 software (Digital Surf, Besançon, France).

The parameters of the surface texture are used primarily for a quantitative assessment of the condition of the surface analysed [5,30]. Texture parameters include, inter alia, amplitude parameters, and the most important of them according to ISO 25178-2:2012 are *Sa*, *Sq* and *Sz*. From a statistical point of view, *Sa* is interpreted as the arithmetic mean height deviation of the texture elements within the sampling area, *Sq* determines the square mean deviation of this elements and *Sz* defines the maximum height of the vertices within the sampling area.

Statistical analysis of measurements was carried out using the Statistica 13 software (StatSoft, Kraków, Poland).

The plan of investigations was developed based on the method named ‘parameter space investigation’ (PSI) that allows planning experiments while minimizing the number of research points [31]. The research points are located in a multidimensional space so that their projections on the X_1_ and X_2_ axes are equally spaced to each other (Figure 1). Point coordinates in the range of X_min_ = 0 and X_max_ = 1 are presented in Table 1.

## 3. Results and Discussion

### 3.1. Chip Shaping

Chip shapes after finishing turning under cooling conditions tested are shown in Figure 2 for minimum surface roughness (*v*_c_~280 m/min and *f*~0.094 mm/rev), medium surface roughness (*v*_c_~456 m/min and *f*~0.27 mm/rev) and maximum surface roughness (*v*_c_~194 m/min and *f*~0.36 mm/rev).

When turning with *v*_c_~280 m/min and *f*~0.094 mm/rev, long and snarling chips were recorded under the cooling conditions tested. Michailidis [32] found that such chips can entangle a cutting tool, leading to its faster wear and at the same time drawing the machined surface.

When turning with *v*_c_~456 m/min and *f*~0.27 mm/rev, short spiral chips can be observed for all investigated cooling conditions, which is desirable for industrial use.

When turning with *v*_c_~194 m/min and *f*~0.36 mm/rev under dry, wet and MQL machining conditions, long spiral chips were obtained and for machining with MQL+EP long screwed chips were observed.

### 3.2. The Turning Conditions Influence on the Surface Texture Amplitude Parameters

The software used made it possible to determine statistical models (Table 2) and graphs of the relationship between *Sa* (Figure 3), *Sz* (Figure 4) and *Sq* (Figure 5) values and feed and cutting speed.

It was found that under dry and wet machining conditions the feed has the greatest influence on the *Sa*, *Sz* and *Sq* texture parameters. The direct effect of the cutting speed and the mutual influence of the feed and cutting speed have only a slight effect on parameters tested. Under MQL and MQL+EP machining conditions, *Sa*, *Sz* and *Sq* values are affected by a feed, cutting speed and their mutual interaction.

During dry and wet turning, lower *Sa*, *Sz* and *Sq* values were registered in conditions of lower feeds and cutting speeds. When turning under MQL and MQL+EP conditions, lower *Sa*, *Sz* and *Sq* values depended on the mutual interaction of the feed and cutting speed.

### 3.3. The Efficiency of Cooling Methods

Figure 6 shows the areas of minimum (0.4–0.8 μm), middle (0.8–1.3 μm) and maximum (1.3–15.0 μm) *Sa* values obtained under dry, wet, MQL and MQL+EP conditions.

When dry machining, minimum *Sa* was registered for lower *f* and medium and higher *v*_c_ values, middle *Sa*—for medium *f* and medium and higher *v*_c_ values, maximum *Sa*—for higher *f* and lower *v*_c_ values. Under wet machining conditions, minimum *Sa* was obtained for lower *f* and medium and higher *v*_c_, middle values—for medium and higher *f* with all *v*_c_, maximum *Sa*—for higher *f* and *v*_c_. Under MQL conditions, minimum *Sa* values were registered for lower *f*, medium and higher cutting speeds, middle values—for medium and higher feeds in the investigated cutting speed range, maximum values—for medium feeds and cutting speeds and also for lower feeds and higher cutting speeds. Under MQL+EP conditions, minimum surface roughness values were obtained for lower feeds and medium cutting speeds, as well as for medium feeds and cutting speeds, middle values—for medium feeds and higher cutting speeds, maximum values—for medium feeds and cutting speeds, as well as for higher feeds and lower cutting speeds.

Bagaber and Yusoff [33] concluded that dry machining is optimal for clean production. To reduce high cutting temperatures, machining fluids are usually used. They have a negative impact on human health and the environment. Therefore, percentage changes in the roughness parameters *Sa*, *Sz* and *Sq* obtained under wet machining and under dry machining conditions were compared.

The main dividing line (black dotted line) of the *Sa* value changes was registered with *f*~0.2 mm/rev (Figure 7). During wet machining compared to dry machining the *Sa* parameter decreased by ~8–26% or increased by ~20–100%. For MQL machining in comparison with dry machining, the *Sa* parameter was reduced by ~18–43% or increased by ~22–541%. Cutting under MQL+EP conditions in comparison with dry cutting reduced the *Sa* value by ~3–48% or increased it by ~38–690%.

In the case of changes in the *Sz* value, no similar dividing line was registered for the range of machining parameters used (Figure 8). During wet turning in comparison with dry machining, the *Sz* parameter decreased by ~12–40% and increased by ~5–132%. For MQL machining in comparison with dry turning the *Sz* parameter decreased by ~39–48% or increased by ~6–271%. When machining under MQL+EP conditions, the *Sz* parameter decreased by ~1–38% compared to dry turning or increased by ~14–500%.

The main dividing line (black dotted line) of the *Sq* value changes was registered with *f*~0.2 mm/rev (Figure 9). During wet machining compared to dry machining the *Sq* parameter decreased by ~5–21% or increased by ~23–100%. For MQL turning in contrast to dry turning, the *Sq* parameter was reduced by ~20–40% or increased by ~26–523%. Cutting under MQL+EP conditions in comparison with dry cutting reduced the *Sq* value by ~2–43% or increased it by ~43–661%.

For the stainless steel tested, in particular in test points 5–7 of the Table 1, significant percentage discrepancies were obtained for the parameters *Sa*, *Sz* and *Sq* depending on the processing conditions. These changes may be result from poor machinability of stainless steels and may be associated, among others, with a strong tendency to deformation reinforcement or build-up-edge formation [34].

### 3.4. The Topography of Surfaces Tested

It is known that surface and subsurface properties depend in particular on surface topography. Kaynak et al. [35] have noted that a more favourable surface topography can ensure a longer use of the product.

Figure 10 shows three-dimensional images and contour maps of the surface after turning with *v_c_*~280 m/min and *f*~0.094 mm/rev under different cutting conditions. After MQL machining, irregularly distributed peaks, some of them of significant height, were observed on the surface. In the case of other cooling methods, unfavourable feed traces in the form of irregular peaks and pits were observed on the surface. The changes observed on the surface are the result of plastic deformation of the material.

Figure 11 shows three-dimensional images and contour maps of the surface after turning with *v_c_*~456 m/min and *f*~0.27 mm/rev. Feed traces were observed on the surface, that are typical of turning. After dry machining, the surface had irregular pits, narrow and high peaks. After wet machining, irregular pits and wide peaks were observed on the surface. After MQL machining, narrow and high ridges with irregular peaks and irregular and deep pits were found on the surface. After processing with MQL+EP, irregular pits and ridges were visible on the surface.

Figure 12 shows three-dimensional images and contour maps of the surface after turning with *v_c_*~195 m/min and *f*~0.36 mm/rev. Feed traces were observed on the surface, that are typical of turning. After dry processing, narrow and thin ridges and wide pits were formed on the surface. After wet processing, ridges and irregular pits were visible on the surface. After near-dry processing, very high ridges and very deep pits were observed on the surface.

Typical texture irregularities on the surface of the machine, namely peaks on 3D images and pits on contour maps, are marked in Figure 10, Figure 11 and Figure 12 with black dotted lines.

It can be concluded on the basis of surface topography analysed that an increase in feed causes an increase in the height of ridges on the surface of 17-4 PH stainless steel regardless of the cooling conditions.

### 3.5. Material Ratio Curves

Based on the shape of the material ratio curve, it was possible to evaluate the tribological behaviour of the surface under analysis. The so-called progressive material ratio curve indicates that the surface analysed has rounded peaks of roughness profiles and will be more resistant to wear in comparison with a digressive shape of these curves with sharp peaks.

Figure 13 shows the material ratio curves and distributions of height of micro-peaks on the surface after finish machining with cooling conditions used for the minimum *Sa* values (*v_c_*~280 m/min and *f*~0.094 mm/rev), medium *Sa* values (*v_c_*~456 m/min and *f*~0.27 mm/rev) and maximum *Sa* values (*v_c_*~194 m/min and *f*~0.36 mm/rev).

For minimum *Sa* (0.4–0.8 μm) under dry, wet and MQL+EP conditions, an anisotropic mixed character of the surface peaks was observed. Whereas, for MQL conditions an anisotropic mixed character with a very large participation of random component was seen.

For medium *Sa* (0.8–1.3 μm) under dry, wet and MQL+EP cooling conditions, an anisotropic cyclical character of the surface peaks was observed, and for MQL machining an anisotropic mixed character with a high participation of random component was registered.

For maximum *Sa* (1.3–15.0 μm), an anisotropic cyclical determined character of the surface peaks was registered for all conditions tested.

The obtained characteristics of the material ratio curves and distributions of peak heights of stainless steel tested are typical for finishing turning, as Oczoś and Liubimov described [36].

### 3.6. Surface Texture Isotropy

The isotropy of the surface topography indicates the stability of its structure in all directions. In other words, it is a generally symmetrical surface texture according to all possible axes of symmetry.

Figure 14 presents surface structure features after finish cutting under the analysed cooling conditions for minimum roughness (*v*_c_~280 m/min and *f*~0.094 mm/rev), medium roughness (*v*_c_~456 m/min and *f*~0.27 mm/rev) and maximum roughness (*v*_c_~194 m/min and *f*~0.36 mm/rev).

In the range of *Sa* = 0.4–0.8 μm, the surface isotropy of ~20% was registered after dry and wet cutting. When machining under MQL conditions, the isotropy of the surface was the highest (~60%), whereas under MQL+EP conditions it was the lowest (~3%). In the range of *Sa* = 0.8–1.3 μm and *Sa* = 1.3–15.0 μm, a small, similar surface isotropy ~6–8% and ~9–10%, respectively, was observed under all cooling conditions investigated. Therefore, it can be stated that the feed and cooling conditions influence the isotropic properties of the surface, as confirmed by Krolczyk et al. [20].

## 4. Conclusions

This paper presents an analysis of changes in the 17-4PH steel chip shapes and surface topography parameters under different machining conditions in a wide feed and cutting speed ranges. Selected amplitude parameters of the surface topography, curves of material ratio and distribution of roughness micro-peaks as well as the efficiency of cooling methods used were investigated. The analysis of the research results revealed that:Under dry, wet, MQL and MQL+EP conditions, the favourable short spiral chips were registered for *v_c_*~456 m/min and *f*~0.27 mm/rev.When dry and wet turning, lower *Sa*, *Sz* and *Sq* surface texture parameters occur in lower feeds and in the wide cutting speed range, while for MQL and MQL+EP conditions the location of the area of minimum values of *Sa*, *Sz* and *Sq* depends both on the feed and cutting speed, as well as on their mutual interaction.The application of the PSI method allowed an effective analysis of the influence of a wide range of factors studied on the surface texture parameters, including both direct and mutual effects.Compared to dry machining, the *Sa* surface texture parameter for wet machining was reduced by ~26%, MQL by ~43% and MQL+EP by ~48%; *Sz* parameter for wet machining by ~40%, MQL by ~48% and MQL+EP by ~38% and *Sq* parameter for wet machining by ~21%, MQL by ~40% and MQL+EP by ~43%.In the case of minimum *Sa* values under dry, wet and MQL+EP cutting irregular small feed traces in the form of pits and peaks are observed, but under the MQL conditions irregularly distributed peaks are formed on the surface, some of them of a significant height. In the case of medium and maximum *Sa* values, traces in the form of deep pits and high peaks typical for turning feed effect were observed under all cooling conditions.Under all cooling conditions tested in the range *Sa* = 0.4–0.8 μm the surface texture has a mixed anisotropic character. An anisotropic periodic character and an anisotropic mixed character of the surface texture are observed under dry, wet and MQL+EP conditions in the range *Sa*= 0.8–1.3 μm and under MQL conditions, respectively. In the range *Sa* = 1.3–15.0 μm under all the examined conditions an anisotropic periodically determined character of the surface tested is registered.In the range *Sa* = 0.4–0.8 μm under dry and wet cutting, the ~20% surface isotropy, under MQL conditions ~60%, and under MQL+EP conditions of ~3% were registered. In the range *Sa* = 0.8–15.0 μm, the surface isotropy in the range from ~6% to ~10% was found under all cooling conditions studied.

## Figures and Tables

**Figure 1 materials-13-02188-f001:**
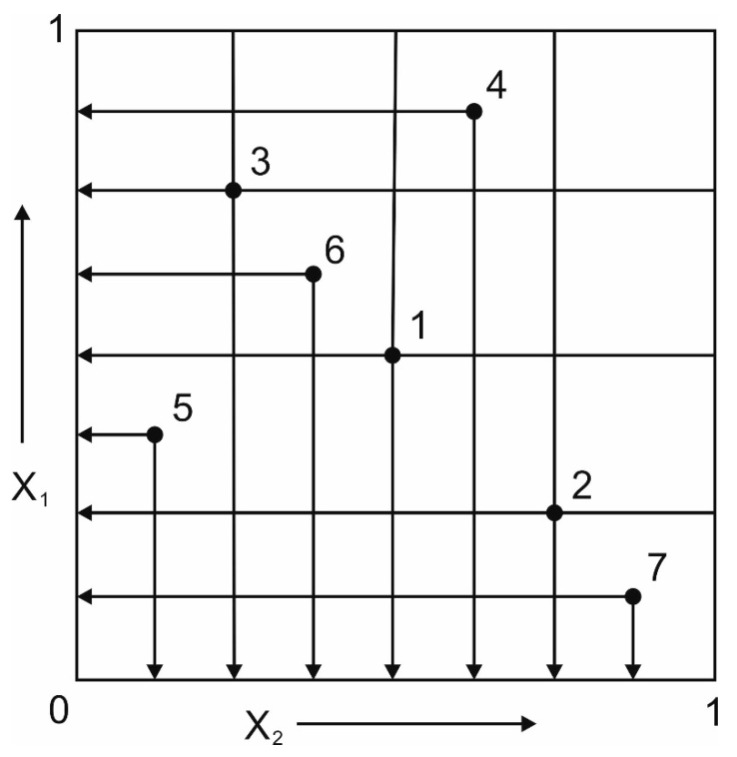
Location of research points on the X_1_ and X_2_ axes.

**Figure 2 materials-13-02188-f002:**
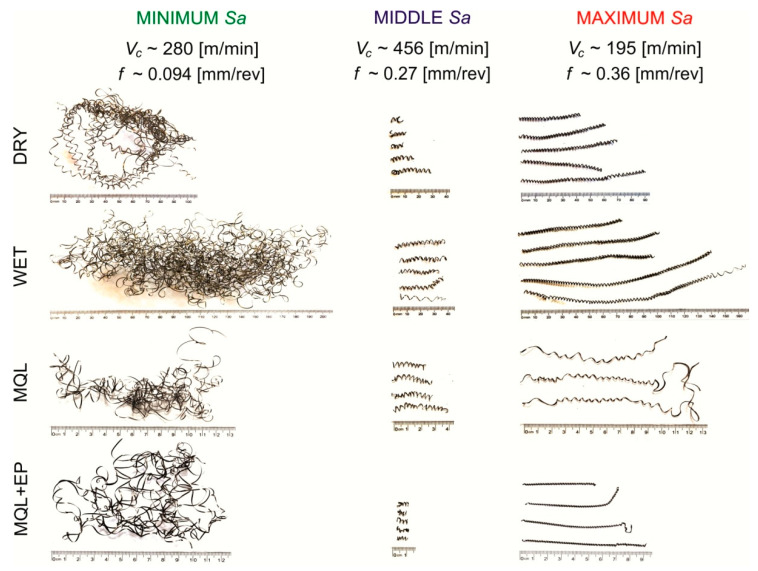
Chip shapes after finish turning of the steel tested.

**Figure 3 materials-13-02188-f003:**
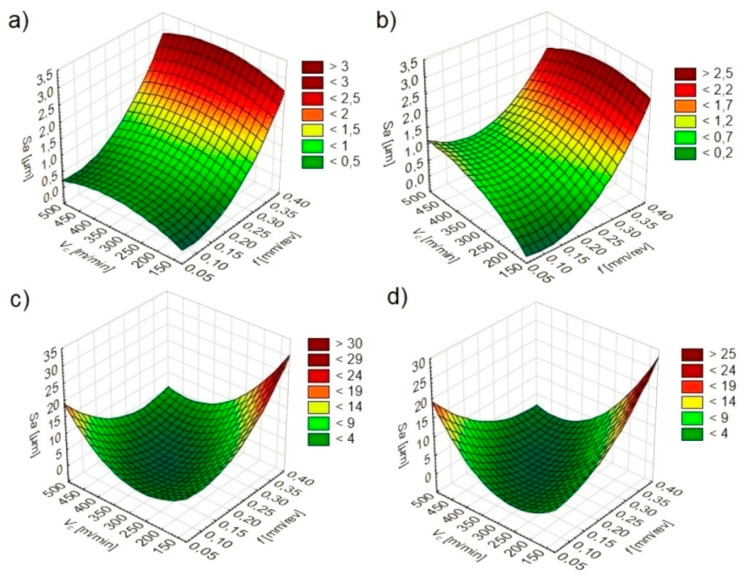
*Sa* dependence on cutting parameters under: (**a**) dry; (**b**) wet; (**c**) MQL; (**d**) MQL+EP conditions.

**Figure 4 materials-13-02188-f004:**
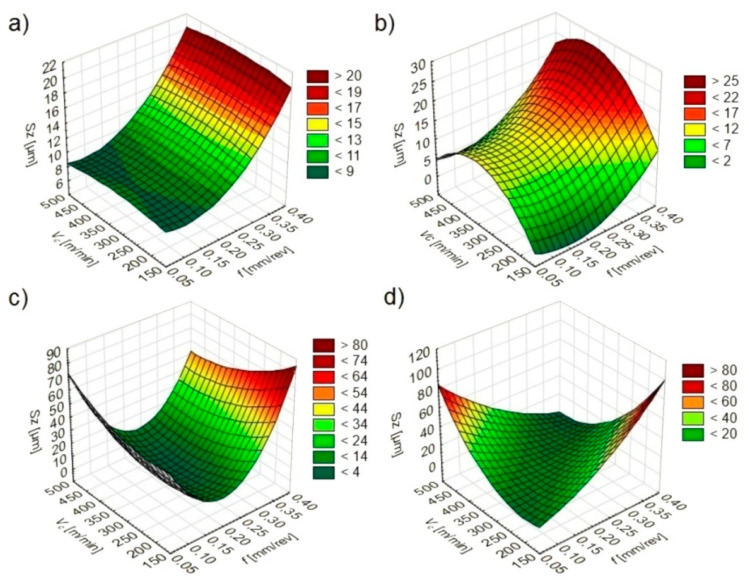
*Sz* dependence on cutting parameters under: (**a**) dry; (**b**) wet; (**c**) MQL; (**d**) MQL+EP conditions.

**Figure 5 materials-13-02188-f005:**
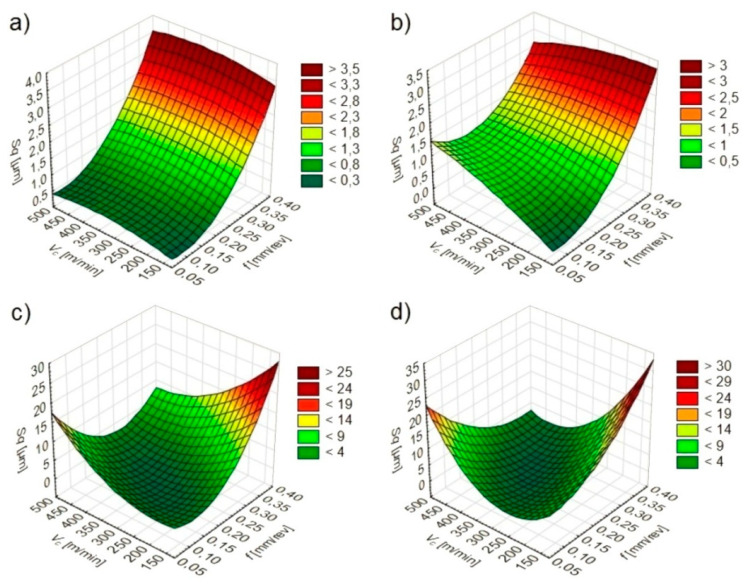
*Sq* dependence on cutting parameters under: (**a**) dry; (**b**) wet; (**c**) MQL; (**d**) MQL+EP conditions.

**Figure 6 materials-13-02188-f006:**
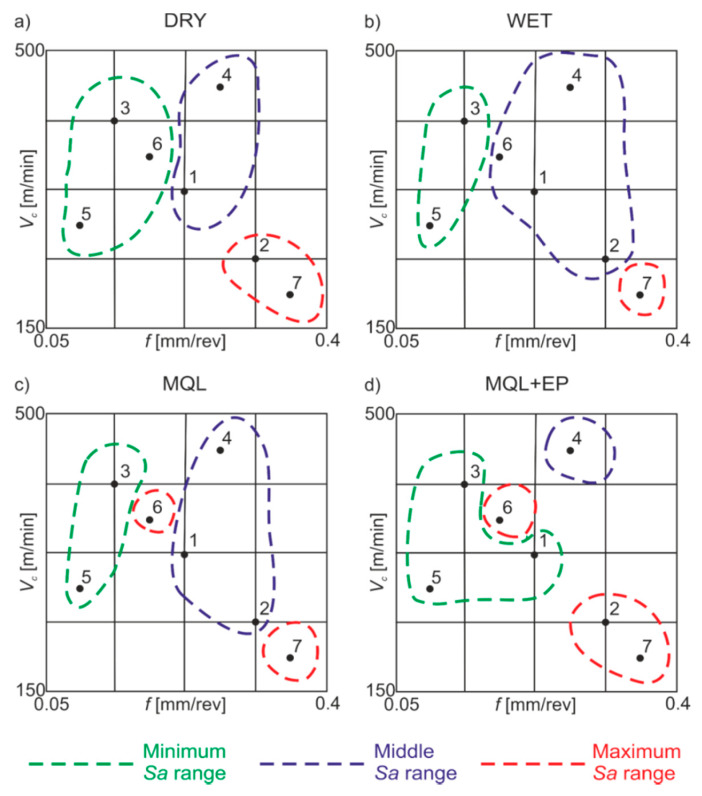
Areas of minimum, medium and maximum *Sa* values of the surface under: (**a**) dry; (**b**) wet; (**c**) MQL; (**d**) MQL+EP conditions.

**Figure 7 materials-13-02188-f007:**
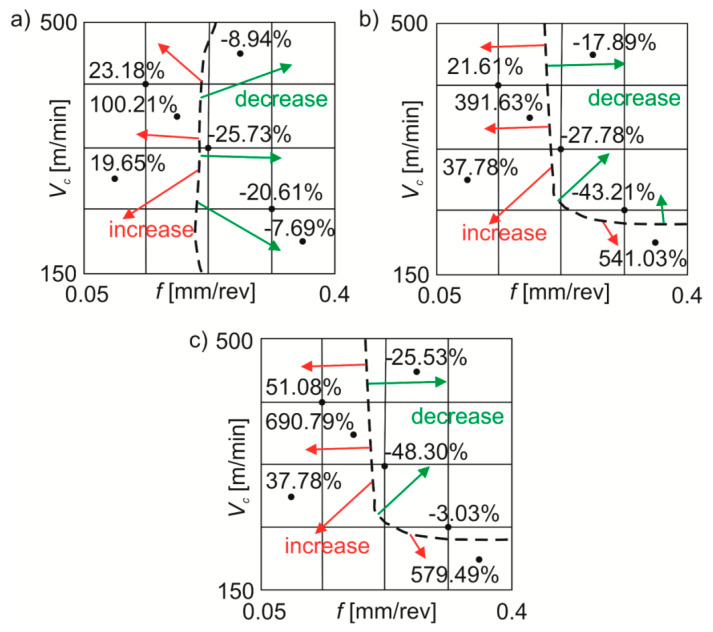
Average percentage changes in the *Sa* roughness parameter under: (**a**) wet turning; (**b**) MQL; (**c**) MQL+EP conditions in comparison with dry turning.

**Figure 8 materials-13-02188-f008:**
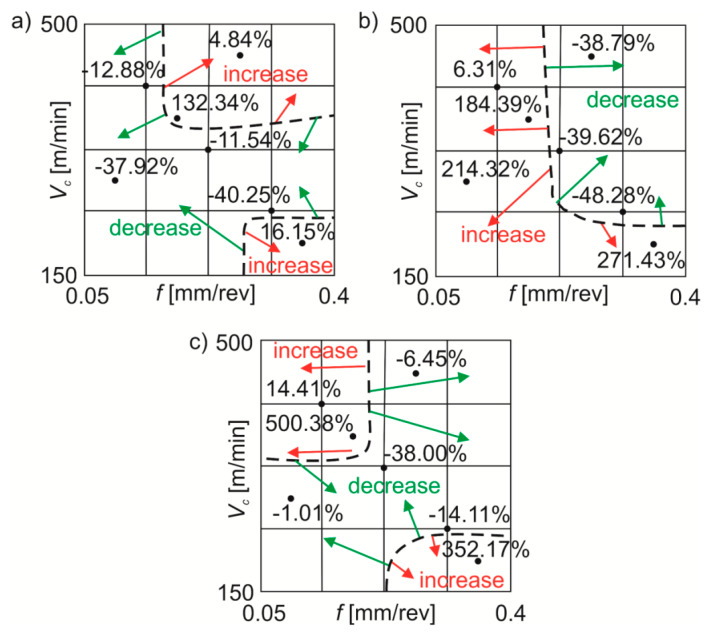
Average percentage changes in the *Sz* roughness parameter under: (**a**) wet turning; (**b**) MQL; (**c**) MQL+EP conditions in comparison with dry turning.

**Figure 9 materials-13-02188-f009:**
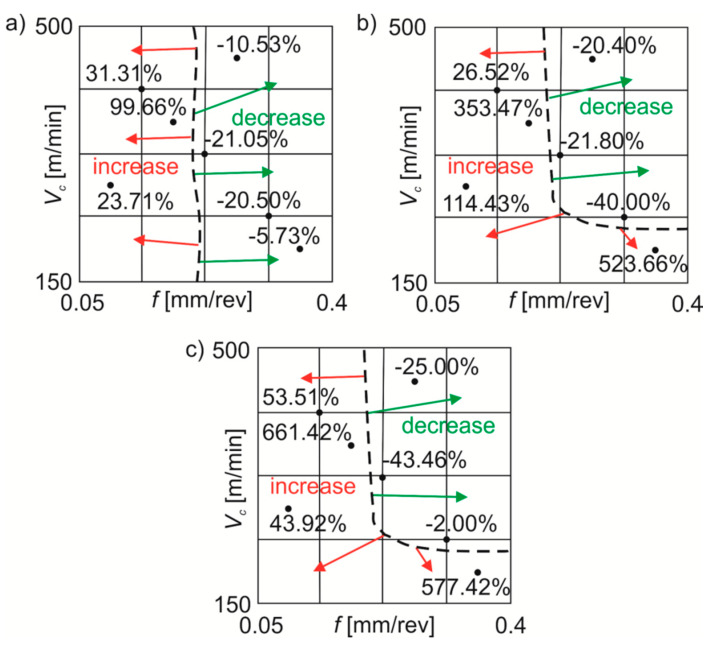
Average percentage changes in the *Sq* roughness parameter under: (**a**) wet turning; (**b**) MQL; (**c**) MQL+EP conditions in comparison with dry turning.

**Figure 10 materials-13-02188-f010:**
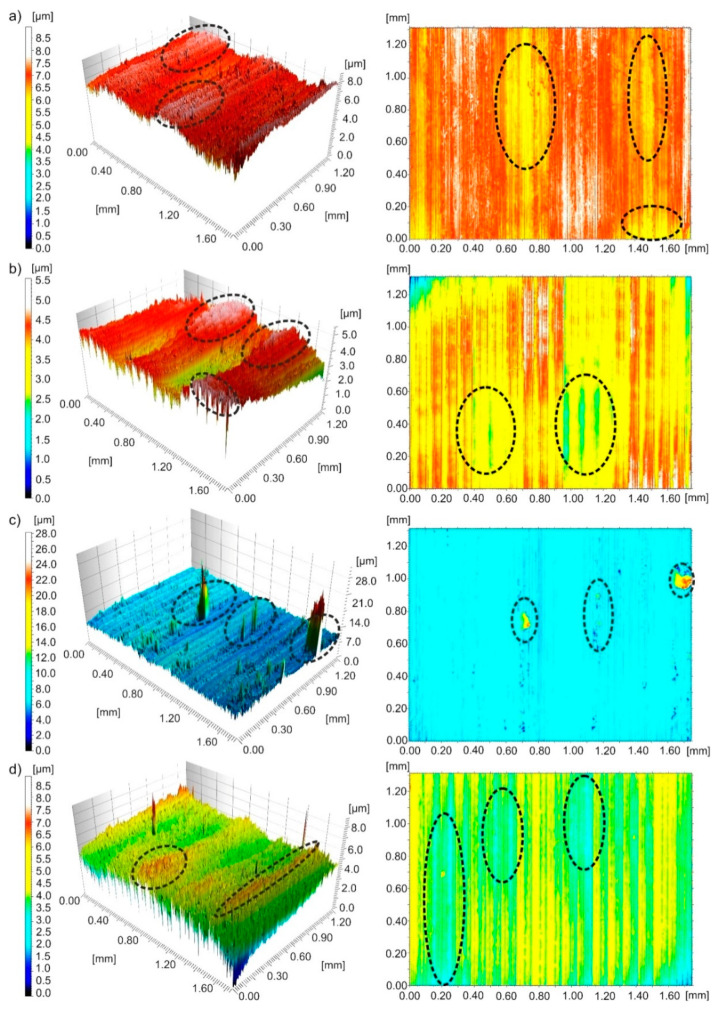
Three-dimensional images and contour maps of the surface machined with a speed of *v_c_*~280 m/min and feed of *f*~0.094 mm/rev obtained under processing conditions: (**a**) dry; (**b**) wet; (**c**) MQL; (**d**) MQL+EP.

**Figure 11 materials-13-02188-f011:**
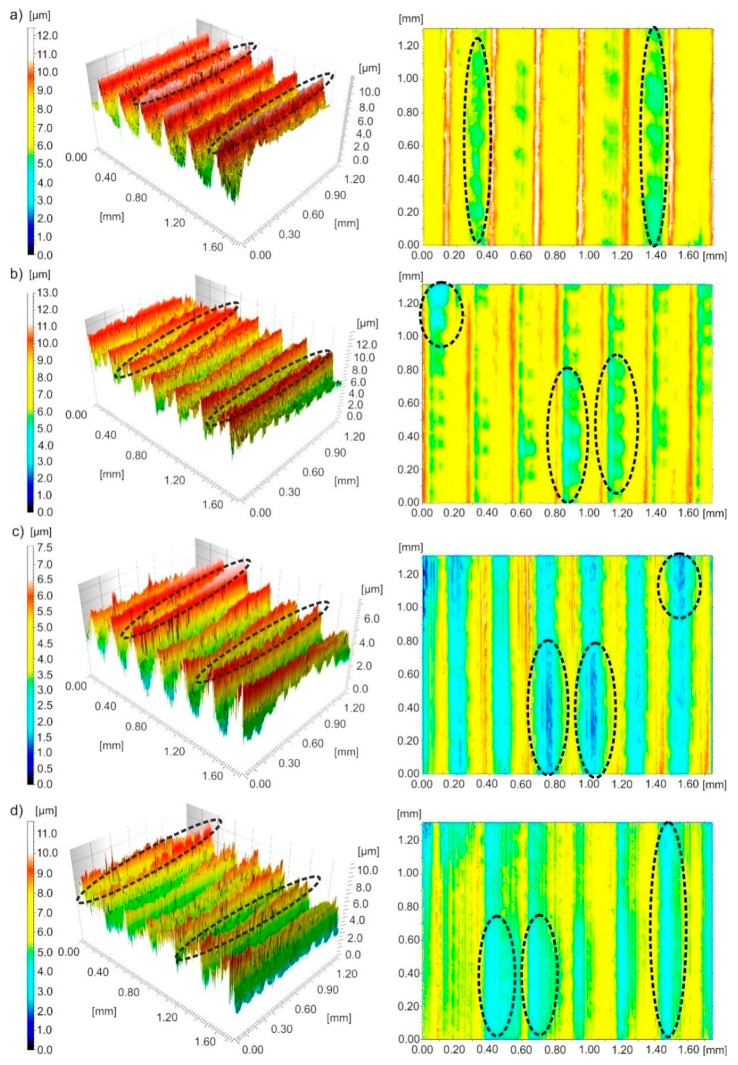
Three-dimensional images and contour maps of the surface machined with a speed of *v_c_*~456 m/min and feed of *f*~0.27 mm/rev obtained under processing conditions: (**a**) dry; (**b**) wet; (**c**) MQL; (**d**) MQL+EP.

**Figure 12 materials-13-02188-f012:**
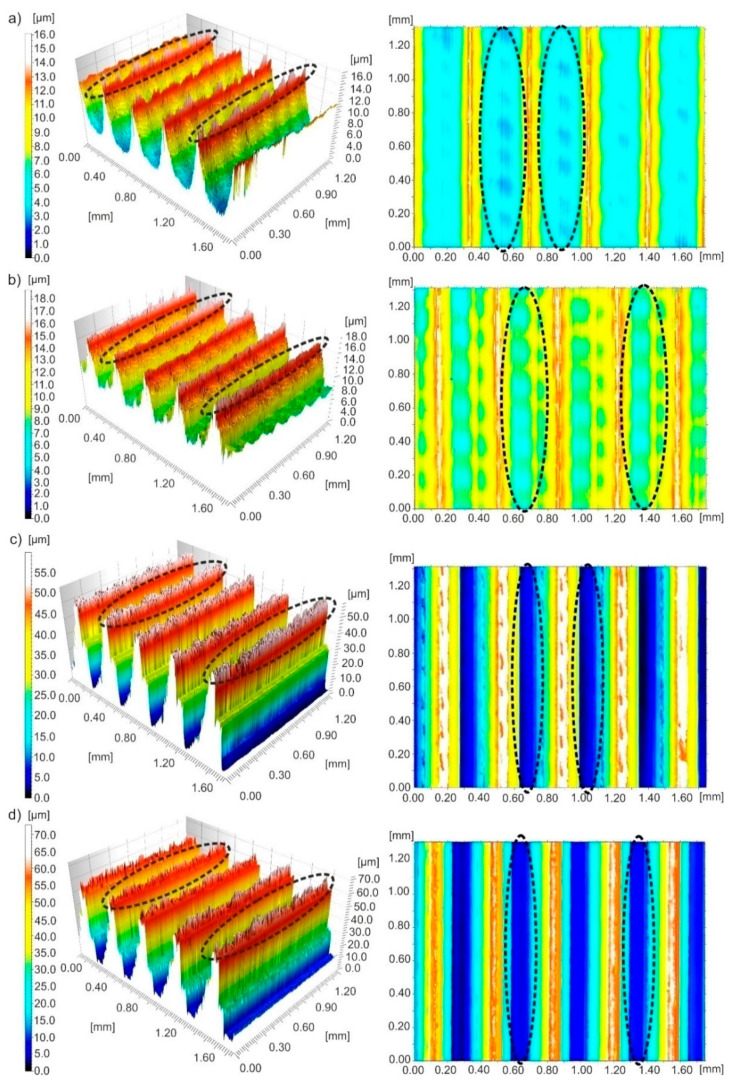
Three-dimensional images and contour maps of the surface machined with a speed of *v_c_*~195 m/min and feed of *f*~0.36 mm/rev obtained under processing conditions: (**a**) dry; (**b**) wet; (**c**) MQL; (**d**) MQL+EP.

**Figure 13 materials-13-02188-f013:**
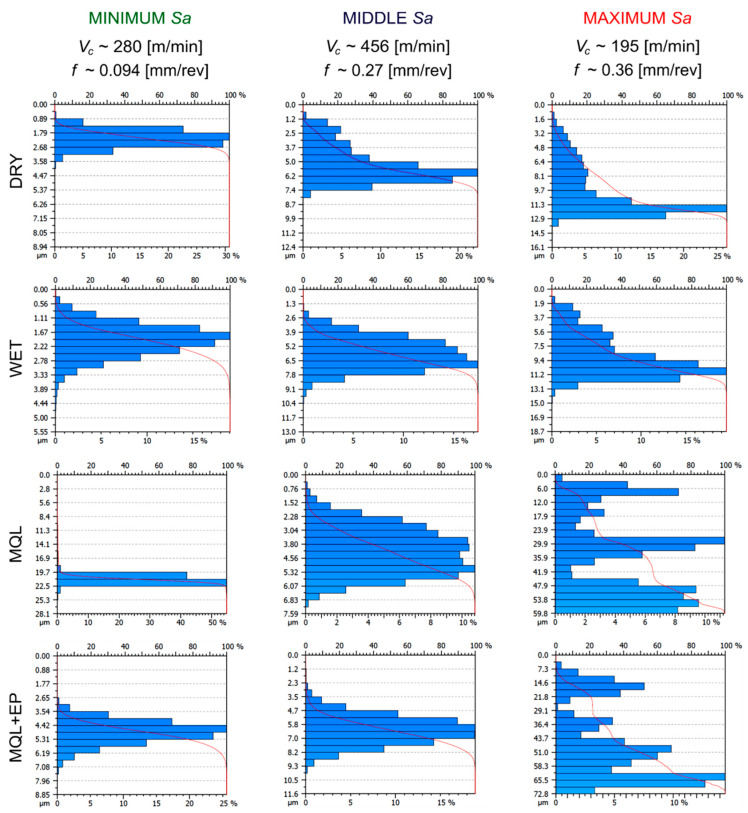
Material ratio curves and distributions of peak heights after finish turning of the steel tested.

**Figure 14 materials-13-02188-f014:**
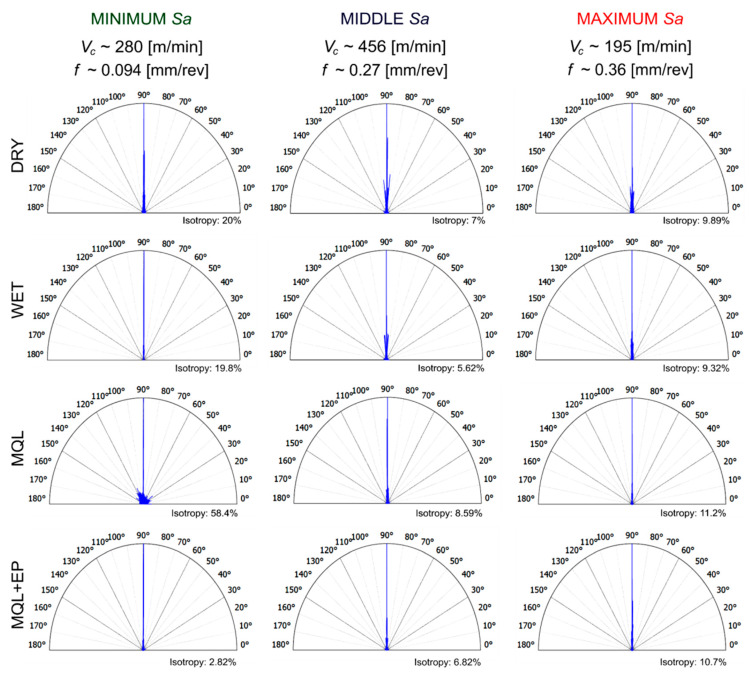
Surface structure direction after finish turning of the steel tested.

**Table 1 materials-13-02188-t001:** Point coordinates.

Variables	Test Points Coordinates						
	1	2	3	4	5	6	7
X_1_	0.5000	0.2500	0.7500	0.8750	0.3750	0.6250	0.1250
X_2_	0.5000	0.7500	0.2500	0.6250	0.1250	0.3750	0.8750

**Table 2 materials-13-02188-t002:** Statistical models for *Sa*, *Sz* and *Sq* calculations after finish processing steel tested under different cooling conditions.

**Dry Cutting**
*Sa* = −0.6872 − 4.4729*f* + 0.0076*V*_c_ + 27.1238*f*^2^ − 0.0003*fV*_c_ − 1.0873·10^−5^*V*_c_^2^
*Sz* = 8.0803 − 26.2563*f* + 0.014*V*_c_ + 129.25*f*^2^ + 0.0035*fV*_c_ − 2.312(E − 5)*V*_c_^2^ *Sq* = 0.0457 − 5.3189*f* + 0.0039*V*_c_ + 31.3739*f*^2^ + 0.0003*fV*_c_ − 5.7246(E − 6)*V*_c_^2^
**Wet Cutting**
*Sa* = −1.7266 − 3.9803*f* + 0.0131*V*_c_ + 32.076*f*^2^ − 0.0136*fV*_c_ − 1.2922E − 5*V*_c_^2^
*Sz*= −26.0225 − 70.2708*f* + 0.229*V*_c_ + 213.7683*f*^2^ + 0.0489*fV*_c_ − 0.0003*V*_c_^2^ *Sq* = −0.8632 − 2.0673*f* + 0.0079*V*_c_ + 30.6233*f*^2^ − 0.0168*fV*_c_ − 4.7939(E − 6)*V*_c_^2^
**MQL Cutting**
*Sa* = 21.3533 − 7.3691*f* − 0.1275*V*_c_ + 233.9231*f*^2^ − 0.2742*fV*_c_ + 0.0003*V*_c_^2^
*Sz* = 85.7288 − 602.4137*f* − 0.0937*V*_c_ + 1726.6324*f* ^2^ − 0.4679*fV*_c_ + 0.0003*V*_c_^2^ *Sq* = 8.0883 − 38.527*f* − 0.0372*V*_c_ + 318.9039*f*^2^ − 0.2655*fV*_c_ + 0.0001*V*_c_^2^
**MQL+EP Cutting**
*Sa* = 17.8488 + 25.895*f* − 0.1264*V*_c_ + 163.9226*f*^2^ − 0.298*fV*_c_ + 0.0003*V*_c_^2^
*Sz* = 26.4998 + 294.6575*f* − 0.2883*V*_c_ + 403.2388*f*^2^ − 1.4351*fV*_c_ + 0.0009*V*_c_^2^ *Sq* = 21.3113 + 30.0252*f* − 0.1502*V*_c_ + 195.6086*f*^2^ − 0.3528*fV*_c_ + 0.0003*V*_c_^2^

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
