# Peer review of "On the Chip Shaping and Surface Topography When Finish Cutting 17-4 PH Precipitation-Hardening Stainless Steel under Near-Dry Cutting Conditions"

_materials, 2020, doi:10.3390/ma13092188_

Round 1

Reviewer 1 Report

The paper brings interesting results about the machining of the 17-4 PH stainless steel. The paper needs some modification before it could be published:

The title should be modified, because 17-4 PH steel is not used for biomedical use only. There are also many applications in chemical industry and other branches.

The discussion of the results is very poor and there is almost no comparison of the results with known findings. At presents state, the paper looks rather like a technical report. I recommend to create a separate "Discussion" chapter. The improvement of the discussion would dramatically increase the scientific impact of the paper.

Author Response

Detailed Response to Reviewer # 1

Thank you for your comments concerning our manuscript entitled after correcting as “On the chip shaping and surface topography when finish cutting 17-4 PH precipitation-hardening stainless steel under near-dry cutting conditions” (Manuscript ID: materials-783420). Those comments are all valuable and very helpful for revising and improving our paper, as well as the important guiding significance to our researches. We have tried our best to revise our manuscript according to the comments, and the revised contexts are highlighted with a yellow background in revision.

The main corrections in the paper and the responds to the reviewer’s comments are as flowing:

1 - The title should be modified, because 17-4 PH steel is not used for biomedical use only. There are also many applications in chemical industry and other branches.

Answer: The authors thank the reviewer for his valuable remark. The title was corrected.

2 - The discussion of the results is very poor and there is almost no comparison of the results with known findings. At presents state, the paper looks rather like a technical report. I recommend to create a separate "Discussion" chapter. The improvement of the discussion would dramatically increase the scientific impact of the paper.

Answer: The authors thank the reviewer for his valuable remark. The authors in the section "Results and discussion" added the information related to the results of other researchers.

Reviewer 2 Report

Some statements in this manuscript require further justification, especially,

(1) The abbreviations such as “MQL, MQL+EP, MQCL” should be explained appropriately when they are used in the text for the first time.

(2) The detailed information such as the manufacturer and the country of the equipment used in the research should be presented. E.g. Line 100, Sensofar S Neox 3D optical profilometer.

(3) It is suggested that the Analysis Of Variance (ANOVA) for the statistical models in Table 2 should be given for the significant test of the models.

(4) Form Figures 2~4, it can be seen that the maximum values of the amplitude parameters under MQL and MQL+EP conditions are much higher than that under dry and wet conditions. For example, the maximum Sa under dry and wet conditions in Figure 2 approximately equals to 3μm whereas the maximum Sa under MQL and MQL+EP conditions is approximately 30μm. Please explain these questions.

(4) In the section “3. Result and discussion”, the authors simply listed the results of the experiment without further discussion. No conclusions or suggestions are given for the optimization of cooling methods according to the machined surface topography and roughness parameters.

(5) Please improve the English and engage native proof reader if available. There are some grammar mistakes and the writing should be paid more attention. Some examples are list as follows:

Line 78: “which chemical composition…” should be revised as “whose chemical composition…”.

Line 81: “contained a small volume of austenite…” should be revised as “containing a small volume of austenite…”.

Line 109: “basing the method…” should be revised as “basing on the method…”.

Line 218: “Material rotio curves” should be revised as “Material ratio curves”.

Author Response

Detailed Response to Reviewer # 2

Thank you for your comments concerning our manuscript entitled after correcting as “On the chip shaping and surface topography when finish cutting 17-4 PH precipitation-hardening stainless steel under near-dry cutting conditions” (Manuscript ID: materials-783420). Those comments are all valuable and very helpful for revising and improving our paper, as well as the important guiding significance to our researches. We have tried our best to revise our manuscript according to the comments, and the revised contexts are highlighted with a yellow background in revision.

The main corrections in the paper and the responds to the reviewer’s comments are as flowing:

1 - The abbreviations such as “MQL, MQL+EP, MQCL” should be explained appropriately when they are used in the text for the first time.

Answer: The authors thank the reviewer for his valuable remark. The abbreviations "MQL, MQL+EP, MQCL" have been explained when they appeared in the text at first time.

2 - The detailed information, such as the manufacturer and the country of the equipment used in the research should be presented. E.g. Line 100, Sensofar S Neox 3D optical profilometer.

Answer: The authors thank the reviewer for his valuable remark. Information on the device manufacturer name and the country has been added.

3 - It is suggested that the Analysis Of Variance () for the statistical models in Table 2 should be given for the significant test of the models.

Answer: The authors thank the reviewer for his remark. It was said in the section 2 that statistical analysis of measurements was carried out using the Statistica 13 software. However, it was not ANOVA but Graphical options.

4 - Form Figures 2~4, it can be seen that the maximum values of the amplitude parameters under MQL and MQL+EP conditions are much higher than that under dry and wet conditions. For example, the maximum Sa under dry and wet conditions in Figure 2 approximately equals to 3μm whereas the maximum Sa under MQL and MQL+EP conditions is approximately 30μm. Please explain these questions.

Answer: The authors thank the reviewer for his valuable remark. These changes result from the data obtained in the point No 7 of the PSI method under near-dry cutting conditions. At this point, with Vc~195 m/min and f~0.36 mm/rev, a significant increase in the amplitude parameters examined was registered compared to dry and wet cutting. In section 3.2 “The efficiency of cooling methods” Figs 6-8 shows the average percentage changes in the Sa, Sz, Sq roughness parameter under: (a) wet turning; (b) MQL; (c) MQL+EP conditions in comparison with dry turning. Please note that in comparison with dry cutting, the near-dry cutting has achieved changes up to ~600%.

5 - In the section “3. Result and discussion”, the authors simply listed the results of the experiment without further discussion. No conclusions or suggestions are given for the optimization of cooling methods according to the machined surface topography and roughness parameters.

Answer: The authors thank the reviewer for his valuable remark. The research results are additionally analysed in section 3 and compared it to the results obtained by other researchers.

6 - Please improve the English and engage native proof reader if available. There are some grammar mistakes and the writing should be paid more attention. Some examples are list as follows:

Line 78: “which chemical composition…” should be revised as “whose chemical composition…”.

Line 81: “contained a small volume of austenite…” should be revised as “containing a small volume of austenite…”.

Line 109: “basing the method…” should be revised as “basing on the method…”.

Line 218: “Material rotio curves” should be revised as “Material ratio curves”.

Answer: The authors thank the reviewer for his remarks. All indicated language errors have been corrected in the text.

Reviewer 3 Report

The paper entitled „On the surface topography when finish cutting 2 biomedical 17-4 PH precipitation-hardening stainless 3 steel under near-dry cutting conditions“ is very interesting. Before further consideration, the paper should be corrected in accordance with the following points:

  1. Think about changing the title. The “title” is the “initial impressions” of a research article, and it needs to be drafted correctly.
  2. Don't use abbreviations without explanation. See abstract, introduction. ..
  3. The abstract is too long. A single paragraph of about 200 words maximum. The abstract should be shortened.
  4. Keywords: Use a semicolon.
  5. The authors should probably provide more information in the experimental part. There are missing information about the equipment used – manufacturer, the number of samples tested. Line 101. The measurement results were prepared using specialized software. Which one? Workpiece material and dimensions? Cutting inserts?
  6. 17-4 PH SS is treated as difficult to cut material. Have the authors encountered problems in machining? What kind of chips were obtained? Is it possible to provide some photographs from the experiment?
  7. I kindly ask the authors to mention form where was the material. It would be better to provide information about mechanical properties.
  8. List your figures in consecutive order, i.e. in the order, they appear in the text with the following discussion.

Author Response

Detailed Response to Reviewer # 3

Thank you for your comments concerning our manuscript entitled after correcting as “On the chip shaping and surface topography when finish cutting 17-4 PH precipitation-hardening stainless steel under near-dry cutting conditions” (Manuscript ID: materials-783420). Those comments are all valuable and very helpful for revising and improving our paper, as well as the important guiding significance to our researches. We have tried our best to revise our manuscript according to the comments, and the revised contexts are highlighted with a yellow background in revision.

The main corrections in the paper and the responds to the reviewer’s comments are as flowing:

1 - Think about changing the title. The “title” is the “initial impressions” of a research article, and it needs to be drafted correctly.

Answer: The authors thank the reviewer for his valuable remark. The title was corrected

2 - Don't use abbreviations without explanation. See abstract, introduction.

Answer: The authors thank the reviewer for his valuable remark. All abbreviations have been explained when they appeared in the text at first time.

3 - The abstract is too long. A single paragraph of about 200 words maximum. The abstract should be shortened.

Answer: The authors thank the reviewer for his remark. The Abstract was shortened.

4 - Keywords: Use a semicolon (;).

Answer: The authors thank the reviewer for his valuable remark. Key words were corrected.

5 - The authors should probably provide more information in the experimental part. There are missing information about the equipment used – manufacturer, the number of samples tested. Line 101. The measurement results were prepared using specialized software. Which one? Workpiece material and dimensions? Cutting inserts?

Answer: The authors thank the reviewer for his valuable remark. Information on the device manufacturer name and the country has been added. Information about the turning tool and the insert used in the tests was added. Information on sample dimensions and number of measurements made for each sample has been added.

6 - 17-4 PH SS is treated as difficult to cut material. Have the authors encountered problems in machining? What kind of chips were obtained? Is it possible to provide?

Answer: The authors thank the reviewer for his valuable remark. The information on chip shaping was added to sections 1 and 3. It discusses the problem of machinability of the steel tested and the chip shapes that have been registered. Some photographs from the experiment also ware added.

7 - I kindly ask the authors to mention form where was the material. It would be better to provide information about mechanical properties.

Answer: The authors thank the reviewer for his remark. This material is normalized and can be bought in many firms.

8 - List your figures in consecutive order, i.e. in the order, they appear in the text with the following discussion.

Answer: The authors thank the reviewer for his valuable remark. The figures have been set in the correct order.

Round 2

Reviewer 1 Report

The paper has been corrected thoroughly and all my comments were accepted.

Author Response

Thank you for the acceptance.

Reviewer 2 Report

The surface texture amplitude parameters, the surface topography, the material ratio curves and the surface texture isotropy under different cutting speed, feed and cooling methods (dry, wet, MQL and MQL+EP) were experimentally investigated in this manuscript. The used methodology and obtained results are of interest. However, some statements in this manuscript require further justification, especially,

(1) The abbreviations such as “MQL, MQL+EP, MQCL” should be explained appropriately when they are used in the text for the first time.

(2) The detailed information such as the manufacturer and the country of the equipment used in the research should be presented. E.g. Line 100, Sensofar S Neox 3D optical profilometer.

(3) It is suggested that the Analysis Of Variance (ANOVA) for the statistical models in Table 2 should be given for the significant test of the models.

(4) Form Figures 2~4, it can be seen that the maximum values of the amplitude parameters under MQL and MQL+EP conditions are much higher than that under dry and wet conditions. For example, the maximum Sa under dry and wet conditions in Figure 2 approximately equals to 3μm whereas the maximum Sa under MQL and MQL+EP conditions is approximately 30μm. Please explain these questions.

(4) In the section “3. Result and discussion”, the authors simply listed the results of the experiment without further discussion. No conclusions or suggestions are given for the optimization of cooling methods according to the machined surface topography and roughness parameters.

(5) Please improve the English and engage native proof reader if available. There are some grammar mistakes and the writing should be paid more attention. Some examples are list as follows:

Line 78: “which chemical composition…” should be revised as “whose chemical composition…”.

Line 81: “contained a small volume of austenite…” should be revised as “containing a small volume of austenite…”.

Line 109: “basing the method…” should be revised as “basing on the method…”.

Line 218: “Material rotio curves” should be revised as “Material ratio curves”.

Author Response

Detailed Response to Reviewer # 2 (second correction)

Thank you for your comments concerning our manuscript entitled after correcting as “On the chip shaping and surface topography when finish cutting 17-4 PH precipitation-hardening stainless steel under near-dry cutting conditions” (Manuscript ID: materials-783420). Those comments are all valuable and very helpful for revising and improving our paper, as well as the important guiding significance to our researches. We have tried our best to revise our manuscript according to the comments, and the revised contexts are highlighted with a yellow background in revision.

The main corrections in the paper and the responds to the reviewer’s comments are as follows:

(1) The abbreviations such as “MQL, MQL+EP, MQCL” should be explained appropriately when they are used in the text for the first time.

Answer. All abbreviations were explained in the text.

(2) The detailed information such as the manufacturer and the country of the equipment used in the research should be presented. E.g. Line 100, Sensofar S Neox 3D optical profilometer.

Answer. The detailed information was added in the section 2.

(3) It is suggested that the Analysis Of Variance (ANOVA) for the statistical models in Table 2 should be given for the significant test of the models.

Answer: The authors thank the reviewer for his remark. It was said in the detailed response (first correction) that in the section 2 statistical analysis of measurements was carried out using the Statistica 13 software. However, it was not ANOVA but Graphical options.

(4) Form Figures 2~4, it can be seen that the maximum values of the amplitude parameters under MQL and MQL+EP conditions are much higher than that under dry and wet conditions. For example, the maximum Sa under dry and wet conditions in Figure 2 approximately equals to 3 μm whereas the maximum Sa under MQL and MQL+EP conditions is approximately 30 μm. Please explain these questions.

Answer. Please, see our comments in the detailed response (first correction).

(4) In the section “3. Result and discussion”, the authors simply listed the results of the experiment without further discussion. No conclusions or suggestions are given for the optimization of cooling methods according to the machined surface topography and roughness parameters.

Answer. The authors thank the reviewer for his valuable remark. The research results are additionally analyzed in section 3 and compared it to the results obtained by other researchers.

(5) Please improve the English and engage native proof reader if available. There are some grammar mistakes and the writing should be paid more attention. Some examples are list as follows:

Line 78: “which chemical composition…” should be revised as “whose chemical composition…”.

Line 81: “contained a small volume of austenite…” should be revised as “containing a small volume of austenite…”.

Line 109: “basing the method…” should be revised as “basing on the method…”.

Line 218: “Material rotio curves” should be revised as “Material ratio curves”.

Answer: The authors thank the reviewer for his remarks. All indicated language errors have been corrected in the text.

Reviewer 3 Report

The authors have satisfactorily responded to all my questions and made
the necessary changes to the manuscript.

Author Response

Thank you very much for the acceptance.

Round 3

Reviewer 2 Report

According to the comments of the reviewers and the editor, the authors have revised the manuscript and the revisions are satisfactory. Therefore, I recommend the acceptance of the manuscript in its present form.